# An Immunobridging Study to Evaluate the Neutralizing Antibody Titer in Adults Immunized with Two Doses of Either ChAdOx1-nCov-19 (AstraZeneca) or MVC-COV1901

**DOI:** 10.3390/vaccines10050655

**Published:** 2022-04-21

**Authors:** Josue Antonio Estrada, Chien-Yu Cheng, Shin-Yen Ku, Hui-Chun Hu, Hsiu-Wen Yeh, Yi-Chun Lin, Cheng-Pin Chen, Shu-Hsing Cheng, Robert Janssen, I-Feng Lin

**Affiliations:** 1International and Governmental Affairs, Medigen Vaccine Biologics Corporation, Taipei 114, Taiwan; josueestrada@medigenvac.com; 2Department of Infectious Diseases, Taoyuan General Hospital, Ministry of Health and Welfare, Taoyuan 330, Taiwan; robinw@mail.tygh.gov.tw (Y.-C.L.); jangbin@mail.tygh.gov.tw (C.-P.C.); shcheng@mail.tygh.gov.tw (S.-H.C.); 3Institute of Public Health, School of Medicine, National Yang-Ming Chiao Tung University, Taipei 112, Taiwan; 4Department of Nursing, Taoyuan General Hospital, Ministry of Health and Welfare, Taoyuan 330, Taiwan; yen376@mail.tygh.gov.tw (S.-Y.K.); 00138@mail.tygh.gov.tw (H.-C.H.); vajien@mail.tygh.gov.tw (H.-W.Y.); 5School of Clinical Medicine, National Yang-Ming Chiao Tung University, Taipei 112, Taiwan; 6School of Public Health, Taipei Medical University, Taipei 110, Taiwan; 7Regulatory and Medical Affairs, Dynavax Technologies Corporation, Emeryville, CA 94608, USA; rjanssen@dynavax.com

**Keywords:** immunobridging, MVC-COV1901, ChAdOx nCOV-19, neutralizing antibodies

## Abstract

Rapid development and deployment of vaccines is crucial to control the continuously evolving COVID-19 pandemic. The placebo-controlled phase 3 efficacy trial is still the standard for authorizing vaccines in the majority of the world. However, due to a lack of eligible participants in parts of the world, this has not always been feasible. Recently, the Taiwan Food and Drug Administration, following the consensus of the International Coalition of Medicines Regulatory Authorities (ICMRA), adopted the use of immunobridging studies as acceptable for authorizing COVID-19 vaccines in lieu of efficacy data. Here, we describe a study in which our candidate vaccine, MVC-COV1901, an adjuvanted protein subunit vaccine, has been granted emergency use authorization (EUA) in Taiwan based on a noninferiority immunobridging study. Immunogenicity results from the per protocol immunogenicity (PPI) subset (*n* = 903) from the MVC-COV1901 phase 2 trial were compared with results from 200 subjects who had received an adenovirus vector vaccine, AstraZeneca ChAdOx nCOV-19 (AZD1222), in a separate study. The lower bound of the 95% confidence interval (CI) of the geometric mean titer (GMT) ratio comparing MVC-COV1901 to AZD1222 was 3.4. The lower bound of the 95% CI of the sero-response rate was 95.5%. Both the GMT ratio and sero-response rate exceeded the criteria established by the Taiwan regulatory authority, leading to EUA approval of MVC-COV1901 in Taiwan.

## 1. Introduction

MVC-COV1901 is a protein subunit SARS-CoV-2 vaccine based on the stabilized prefusion spike protein S-2P adjuvanted with CpG 1018 and alum [1]. From 2020 to early 2021, Taiwan was spared from the worst of the pandemic, recording 525 local and imported cases in 2020 and 339 cases in 2021 prior to a local outbreak in May 2021 [2]. As a result, it was not feasible to conduct a placebo-controlled efficacy trial in Taiwan. In response, the Taiwan Food and Drug Administration (TFDA) designed a pathway to EUA for all local vaccine candidates based on immunobridging, which compares the immune response of a vaccine candidate with an approved vaccine [3]. It is assumed that a part of the immune response—such as binding or neutralizing antibodies—will correlate with other important components of the immune response [4]. In the absence of efficacy data, immunobridging can be adopted as an approach to infer the likelihood of a vaccine’s protective effect by translating immunogenicity to vaccine efficacy. The inference is based on the comparison of immunogenicity of a new vaccine with a comparator vaccine with an established protective effect [4].

In June 2021, experts from regulatory authorities around the world convened at a workshop for the future of COVID-19 vaccine development, and a consensus was reached for the use of well-justified and appropriately designed immunobridging studies in place of efficacy studies when they are not feasible [5]. The results of a phase 2 clinical trial for MVC-COV1901 with over 3800 participants allowed the Taiwan regulatory authorities to compare the safety and immunogenicity to the previously approved ChAdOx nCOV-19 (AZD1222) vaccine [3,6]. MVC-COV1901 was approved by the TFDA in July 2021, making it among the first COVID-19 vaccines approved using an immunobridging study prior to the availability of efficacy data [7]. In September 2021, the consensus position has since been taken up by the Access Consortium, which consisted of regulatory authorities from the UK, Australia, Canada, Singapore, and Switzerland, to accept immunobridging studies as sufficient for authorizing COVID-19 vaccines [8,9]. This manuscript provides an example of a COVID-19 vaccine approved with an immunobridging study, an approach that has gradually become recognized by health regulatory authorities worldwide. 

## 2. Methods

### 2.1. Clinical Trials and Sample Population

The MVC-COV1901 phase 2 trial was a prospective, randomized, double-blind, placebo-controlled, and multicenter study to evaluate the safety, tolerability, and immunogenicity of the SARS-CoV-2 vaccine candidate MVC-COV1901 (NCT04695652) [6]. Individuals were enrolled between December 2020 and April 2021 and received two doses of MVC-COV1901 4 weeks apart. The safety population consisted of 3844 subjects ≥ 20 years of age who were generally healthy or with stable pre-existing medical conditions recruited from 11 sites in Taiwan [6]. The per-protocol immunogenicity (PPI) subset consisted of 903 subjects who received two doses of MVC-COV1901 as scheduled in the clinical trial [6]. In a separate study, 200 generally healthy subjects ≥ 20 years of age were recruited among healthcare workers at the Taoyuan General Hospital, Ministry of Health and Welfare, from March to June 2021, and administered two doses of AZD1222 8 weeks apart. In both studies, individuals with previous known or potential exposure to COVID-19 were excluded. Serum for neutralizing antibody titers was collected on the day of vaccination (pre) and 28 days after the second dose of the vaccine (post).

For the MVC-COV1901 vaccine, the sample size was based on lot-to-lot consistency. The estimation was based on two-sided 95% CI [100(1 − 2α)%] calculated using the differences of the post-vaccination GMT of neutralizing antibody titers between a pair of lots. The total sample size and the size of each age subgroup was based on the minimum requirement of the TFDA. At the time of the study, there were no data available for the standard deviation of neutralizing antibody titers from AZD1222; therefore, the sample size for AZD1222 was not justified at the time the protocol was written. The trial protocol and informed consent form were approved by the TFDA and the ethics committees at the participating sites. The institutional review boards included the Chang Gung Medical Foundation, National Taiwan University Hospital, Taipei Veterans General Hospital, Tri-Service General Hospital, Taipei Medical University Hospital, Taipei Municipal Wanfang Hospital, Taoyuan General Hospital Ministry of Health and Welfare, China Medical University Hospital, Changhua Christian Hospital, National Cheng Kung University Hospital, and Kaoshiung Medical University Hospital. The trial was conducted in accordance with the principles of the Declaration of Helsinki and good clinical practice guidelines.

### 2.2. Vaccines

MVC-COV1901 is a subunit vaccine consisting of the SARS-CoV-2 prefusion spike protein (S-2P) adjuvanted with 750 μg CpG 1018 and 375 μg aluminum hydroxide. A standard 0.5 mL dose contains 15 µg of the Spike-2P. The comparator vaccine is ChAdOx nCOV-19 (AZD1222), an adenoviral vector vaccine developed by University of Oxford, Oxford, United Kingdom and AstraZeneca, Cambridge, United Kingdom. Each dose of vaccine is 0.5 mL and contains 5 × 10^10^ viral particles. Both vaccines are delivered intramuscularly in the deltoid.

### 2.3. Immunobridging Study

According to the TFDA, the following criteria were established for a candidate vaccine to be granted EUA in Taiwan [3]:Immunogenicity data: immunobridging study to evaluate the immunogenicity of a locally developed vaccine against a comparator vaccine which has already been approved in Taiwan.Safety data: at least 3000 subjects were required to be tracked for at least one month after the last dose and all subjects to be followed for a median of two months after the last dose.

As AZD1222 was the first COVID-19 vaccine to be approved in Taiwan, it was chosen as the comparator vaccine for which the locally developed vaccines are to be benchmarked [3]. The immunobridging criteria were to meet the following endpoints for serum samples 28 days after the second dose (day 57) in a population under the age of 65 [3,9]:The lower limit of the 95% CI of the geometric mean titer ratio (GMTR) of the prototype strain live virus neutralizing antibodies for the MVC-COV1901 vaccine group to the external control group must be greater than 0.67;The sero-response level (the proportion of subjects whose neutralizing antibody titers against the prototype strain live virus at 28 days after receiving the second dose of the MVC-COVID19 vaccine) was defined as the proportion of subjects with neutralizing antibody titers against the prototype strain live virus at 60% of the reverse cumulative distribution curve for the external control group. The lower limit of the 95% CI for the sero-response rate must be greater than 50%.

### 2.4. Live SARS-CoV-2 Neutralization Assay

Neutralizing antibody titers against the Wuhan prototype SARS-CoV-2 strain were determined by a live virus neutralization assay performed in our phase 2 clinical study. The detection and characterization of neutralizing antibodies were performed by central laboratories using validated live virus neutralization assays. The live SARS-CoV-2 neutralization assay was performed as previously described with wildtype SARS-CoV-2, Taiwan CDC strain number 4 (hCoV-19/Taiwan/4/2020; GISAID accession ID EPI_ISL_411927) [6]. Briefly, the serum samples underwent a total of eight twofold dilutions, starting from a 1:8 dilution to a final dilution of 1:1024. Diluted serum samples were then mixed with an equal volume of 100 TCID_50_ per 50 μL of virus and incubated at 37 °C for 1 h. After incubation, the mixture was added to Vero E6 cells and incubated at 37 °C in a 5% CO_2_ incubator for 4–5 days. The neutralizing titer (NT_50_) was estimated as the reciprocal of the highest dilution capable of inhibiting 50% of the cytopathic effect. The Reed–Muench method was used to calculate the NT_50_. Neutralizing antibody titers were then converted to the WHO Standardized Unit, IU/mL. The conversion was based on the WHO validated NIBSC reference panel. The GMT was derived from the results and converted to international units (IU/mL) [6].

### 2.5. Statistical Analysis

Descriptive statistics are presented for the population’s demographic and baseline characteristics. GMTs were estimated from neutralizing antibody titers measured at 28 days after the second dose of the study intervention. GMTR is calculated as the GMT of MVC-COV1901 group over the GMT of AZ1222 group.

The GMTs are presented with their two-sided 95% CIs. Sensitivity analyses were performed by excluding elderly participants and those with elevated baseline neutralizing antibody levels. The most extreme modification of the dataset excluded participants with a GMT higher than the 67th percentile at day 57. Assessment of the magnitudes of differences in immune response between the two vaccines was conducted using an analysis of covariance (ANCOVA) model. The model included the log-transformed antibody titers at day 57 as the dependent variable, vaccine group (AZD1222 and MVC-COV1901) as an explanatory variable, and adjusted for age, body mass index (BMI), gender, and comorbidity profile. The 95% CI for the adjusted neutralizing antibody titers of each vaccine group were obtained. Then, adjusted GMT and corresponding 95% CI were back-transformed to the original scale. Lastly, to illustrate the strength of the immune response, reverse cumulative distribution (RCD) curves were constructed for data 28 days after the 2nd dose for the AZD1222 and MVC-COV1901 groups. As described by Reed et al. [10], RCD curves are step functions based on the order statistics of the data. The curve begins with a value of 1.0 or 100% at an antibody titer of zero and falls to a value of zero above the highest titer. In the case of ties, the step size becomes the equivalent of the number of tied values times 1/*n* [10].

## 3. Results

### 3.1. Demographics

All participants were of Asian descent. The mean age of the AZD1222 and MVC-COV1901 groups were similar but the MVC-COV1901 group had more elderly (>65 years old) participants (Table 1). The AZD1222 group had more female participants and more participants with comorbidities than the MVC-COV1901 group.

### 3.2. Immunogenicity

In subjects under the age of 64, at 28 days after the second dose, the AZD1222 and MVC-COV1901 groups had GMTs of 186 and 733, respectively (Figure 1). When including subjects 65 years of age and older, the GMTs decreased to 184 and 662 for the AZD1222 and MVC-COV1901 groups, respectively (Figure 1). For the immunobridging comparison, the lower limit of the 95% CI for the GMTR of the prototype strain live virus neutralizing antibodies between MVC-COV1901 and AZD1222 groups was 3.4, which was greater than the requirement of 0.67. The lower limit of the 95% CI for the sero-response rate of the MVC-COV1901 group was 95.5%, which was greater than the requirement of 50%.

Illustrated in Figure 2 are the RCD curves of neutralizing antibody titers. Higher neutralizing antibody titers were observed in the MVC-COV1901 group than in the AZD1222. At the reference point of 60%, AZ1222 recipients had neutralizing antibody titers ≤199.5 IU/mL, which was equivalent to approximately 90% of MVC-COV1901 recipients.

## 4. Sensitivity Analysis

The sensitivity analyses conducted to detect the robustness of GMT results reveal that both AZD1222 and MVC-COV1901 enhanced neutralizing antibody titers in both the subset of the younger individuals (aged 20–64 years) and the overall sample. The GMTR in the younger group (3.89; 95% CI: 3.45, 4.4) was comparable to the GMTR in the total population (3.55; 95% CI: 3.2, 3.97). The subgroup of younger individuals had higher GMTs compared to the overall GMTs for both vaccines. After adjusting for age, sex, BMI, and comorbidity status, the GMTR was similar in the total population and the younger age group (Table 2). Findings are consistent in the sensitivity analysis that excluded GMTs greater than or equal to the 67th percentile at day 57. GMTs are lower in the total population than in the younger age group (Table 3). In this analysis, the adjusted GMTR is also similar between the total population and the younger age group. Subgroup analyses based on gender and comorbidity profile show consistency in estimates across subgroups.

## 5. Discussion

Multiple regulatory agencies now consider that a primary endpoint of neutralizing antibodies induced by an investigational COVID-19 vaccine compared with those of a vaccine authorized based on efficacy is sufficient for approving new COVID-19 vaccines. The decision has become relevant as efficacy studies have become less feasible. Consistent with the recommendation by the International Coalition of Medicines Regulatory Authorities (ICMRA) [11], our study demonstrated superior immunogenicity of MVC-COV1901 to AZD1222 as a predictor of vaccine effectiveness. The regulator’s consortium recommended a non-inferiority design with an active comparator with high efficacy, or superiority for an active comparator with modest efficacy. Based on the efficacy of AZD1222, demonstrating non-inferiority was adequate to gain regulatory approval. Our study sought to replicate the immunogenicity comparison between MVC-COV1901 and AZD1222 conducted by the TFDA through which an EUA was granted to MVC-COV1901. This study was a post hoc comparison of immunogenicity from the vaccines in two different studies, not a randomized, controlled, blinded study. To increase confidence in the results from the regulator’s perspective, a series of sensitivity analyses were conducted. The sensitivity analysis showed that, after omitting the highest 33 percent of neutralizing antibody titers, the GMT and GMTR were consistent.

The neutralizing antibody titers were determined using World Health Organization (WHO)-certified reference standards, International Unit, IU/mL. The use of the standardized unit to report humoral immunogenicity could facilitate future cross-platform or cross-lab comparison. Moreover, since the participants in this study were mainly Asian, reporting our results in IU/mL can facilitate cross-ethnicity comparisons with other studies.

MVC-COV1901 was well-tolerated without observed safety concerns in the phase 2 study [6]. In addition, the V-Watch program, launched by the Taiwan Centers for Disease Control to monitor post-marketing safety, has reported no serious adverse effects for the MVC-COV1901 vaccine. Its database contains data from more than 2 million doses which have been administered as of March 2022. MVC-COV1901 presented a favorable safety profile compared to vaccines utilizing other platforms [12]. In combination with the safety data, results of this immunobridging analysis support the use of protein subunit vaccines as safe and more tolerable options with a robust immune response. Results of studies conducted to test MVC-COV1901 as a booster shot also warrant that the vaccine offers a robust immune response while maintaining a favorable safety profile [13,14]. These findings provide a safe and effective alternative for the completion of primary and booster immunization, which potentially can accelerate vaccination globally, especially in low- and middle-income countries.

Results of the study corroborate earlier findings, which suggest adenoviral vector vaccines generate lower neutralizing antibodies but, not examined in this study, higher T cell responses compared to protein subunit vaccines [15]. For AZD1222, overall vaccine efficacy more than 14 days after the second dose was 66.7% [13]. Levels of neutralizing antibodies can be correlated and used to predict vaccine efficacy [16,17]. Our findings suggest that levels of neutralizing antibodies of two doses of MVC-COV1901 correlate to approximately 90% vaccine efficacy against the prototype strain [18].

The most important limitation of this study is that it was a post hoc comparison of immunogenicity from two different studies. While the two studies were conducted at different times, both were conducted in the same clinical trial site, the AZD1222 study was conducted in one of the sites used in the phase 2 trial by the same investigators, using the same eligibility criteria. For example, those with previous known or potential exposure to COVID-19 were excluded in both studies. Additionally, cell-mediated immunity was not included in the comparative immunogenicity profile. Lastly, other characteristics of interest, such as waning immunity and cross-reactivity against variants of concern (VoCs), were not explored.

The data presented in the study showed that it is reasonably likely that the vaccine efficacy of MVC-COV1901 is similar or superior to that of AZD1222. These data may have the potential to be used in support of further vaccine development and regulatory approvals.

## 6. Conclusions

Immunobridging offers an alternative approach in cases wherein efficacy studies are not feasible. In this post hoc analysis, we have shown that MVC-COV1901, a protein subunit vaccine, elicits a comparable or superior immune response to that of AZD1222. This became the basis of its approval for an EUA in Taiwan. These findings have implications on the further development of vaccines for regulatory approval. In the long term, this approach may help improve the rate of vaccination and availability of vaccines, especially in low- and middle-income countries.

## Figures and Tables

**Figure 1 vaccines-10-00655-f001:**
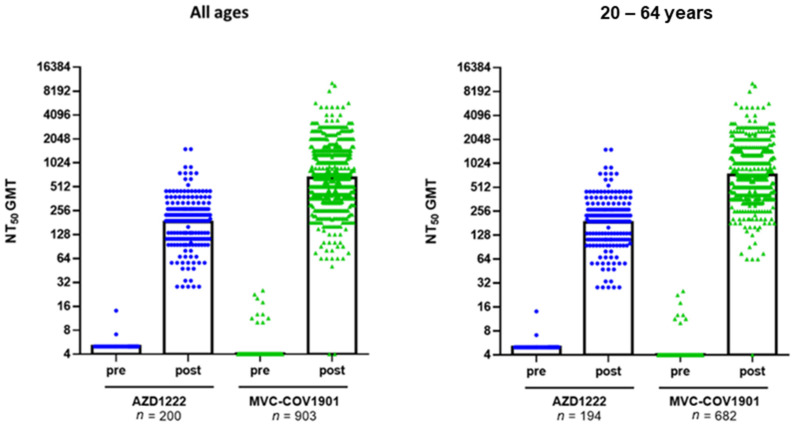
Neutralizing antibody titer in subjects immunized with two doses of either AZD1222 or MVC-COV1901 in all ages (**left**) and ages 20–64 years (**right**). Serum samples were taken before the first vaccination (pre) and 28 days (post) after the second dose of either vaccine and were analyzed in a live SARS-CoV-2 neutralization assay. The results are shown as 50% neutralizing titer (NT_50_), with symbols indicating individual NT_50_ values and the bars indicating the GMT of each group.

**Figure 2 vaccines-10-00655-f002:**
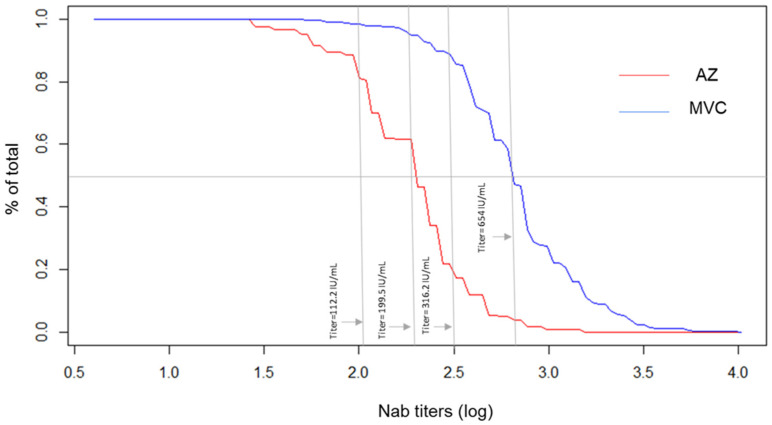
Reverse cumulative distribution curve using log-transformed titers by percent of subjects who had neutralizing antibody titers 28 days following the second dose of AZ1222 and MVC-COV1901.

**Table 1 vaccines-10-00655-t001:** Demographics of the population groups.

Item	<65 Years		All Ages	
	AZD1222	MVC-COV1901	AZD1222	MVC-COV1901
	*n* = 194	PPI Subset *n* = 682	*n* = 200	PPI Subset *n* = 903
**Age (years)**				
Mean (SD)	42.2 (11.1)	38.02 (11.03)	42.9 (11.7)	45.68 (16.64)
Median (IQR)	41(16.75)	37.0 (17.0)	41.5(17.2)	42.0 (32.0)
Min–Max	22.0–64.0	20.0–64.0	22.0–69.0	20.0–87.0
**Gender**				
Male	78 (40.2)	386 (56.6)	80 (40.0)	521 (57.7)
Female	116 (59.8)	296 (43.4)	120 (60.0)	382 (42.3)
**BMI (kg/m^2^)**				
Mean (SD)	25.3 (4.5)	24.9 (4.3)	25.2(4.6)	24.9 (4.1)
Median (IQR)	24.6 (5.97)	24.3 (5.7)	24.5(5.97)	24.4 (5.3)
Min–Max	16.6–39.6	14.3–45.2	17.01–37.5	17.7–36.9
**BMI group**				
<30 kg/m^2^	159 (81.96)	594 (87.1)	798 (88.4)	164 (82.0)
≥30 kg/m^2^	35 (18.04)	88 (12.9)	105 (11.6)	36 (18.0)
**Pre-vaccination neutralizing antibody status**				
Seropositive	2 (1.03)	8 (1.17)	2 (1.0)	10 (1.11)
Seronegative	192 (98.97)	674 (98.83)	198 (99.0)	893 (98.9)
**Comorbidity Category**				
At least one comorbidity	74 (38.1)	89 (13.0)	123 (61.5)	729 (80.7)
No comorbidity	120 (61.9)	593 (87.0)	77 (38.5)	174 (19.3)

**Table 2 vaccines-10-00655-t002:** Neutralizing antibody titers and adjusted GMT ratios in subjects immunized with either two doses of AZD1222 or MVC-COV1901 in all ages and ages 20–64 years at day 57 (28 days after the second dose).

Item	<65 Years			All Ages		
	AZD1222 *n* = 192	MVC-COV1901 (PPI Subset) *n* = 674	*p*-Value (GMT Ratio)	AZD1222 *n* = 198	MVC-COV1901 (PPI Subset) *n* = 893	*p*-Value (GMT Ratio)
**GMT** **(95% CI)**	185.97 (167.3–206.7)	723.6 (683.7–765.8)		184.05 (166.5–204.7)	654.07 (620.9–689.03)	
**GMT Ratio of MVC/AZ** **(95% CI)**	3.89 (3.45–4.4)		<0.0001	3.55 (3.2–3.97)		<0.0001
**Adjusted GMT Ratio * (95% CI)**	3.78 (3.3–4.3)		<0.0001	3.8 (3.4–4.3)		<0.0001

* The GMT ratio was adjusted for age, gender, BMI, and comorbidity profile using general linear models. GMT—geometric mean titer, PPI—per protocol immunogenicity.

**Table 3 vaccines-10-00655-t003:** Sensitivity analysis excluding subjects with a GMT higher than the 67th percentile of neutralizing antibody titer in subjects immunized with either two doses of AZD1222 or MVC-COV1901 in all ages and ages 20–64 years at day 57 (PPI subset).

Item	<65 Years			All Ages		
	AstraZeneca AZD1222 *n* = 192	MVC-COV1901 (PPI Subset) *n* = 458	*p*-Value (GMT Ratio)	AstraZeneca AZD1222 *n* = 198	MVC-COV1901 (PPI Subset) *n* = 635	*p*-Value (GMT Ratio)
**GMT** **(95% CI)**	185.97 (167.3–206.7)	492.6 (470.1–516.1)		184.05 (166.5–204.7)	453.7 (433.8–474.7)	
**GMT Ratio of MVC/AZ** **(95% CI)**	2.65 (2.4~2.97)		<0.0001	2.46 (2.2–2.7)		<0.0001
**Adjusted GMT Ratio *** **(95% CI)**	2.6 (2.3–2.9)		<0.0001	2.62 (2.3–2.9)		<0.0001

* The GMT ratio was adjusted for age, gender, BMI, and comorbidity profile using general linear models. GMT—geometric mean titer, PPI—per protocol immunogenicity.

## Data Availability

The original data presented in the study are available upon reasonable request to the authors.

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
