# Peer review of "An Immunobridging Study to Evaluate the Neutralizing Antibody Titer in Adults Immunized with Two Doses of Either ChAdOx1-nCov-19 (AstraZeneca) or MVC-COV1901"

_vaccines, 2022, doi:10.3390/vaccines10050655_

Round 1

Reviewer 1 Report

In this manuscript, Estrada and colleagues performed a very relevant immunobridging study to compare the neutralizing antibody generated after vaccination with two doses of AstraZeneca and MVC-COV1901 vaccines. Overall, the results are very important for the scientific (and non-scientific) communities, and therefore they should be published.

I only have few suggestions to improve the manuscript:

  • Please explain the concept of immunobridging already in the abstract
  • It would be optimal to introduce the differences between the two types of vaccines (for example in the introduction, and eventually to discuss it in the discussion part)
  • The authors reported that individuals were generally healthy. Did they measure any objective parameter (for example CRP or inflammation in the blood) to define them healthy or not? In addition, were subjects who experienced COVID-19 before the current study excluded?
  • Line 120: “Neutralizing antibody titres against the Wuhan prototype SARS-CoV-2 were determined by SARS-CoV-2 live virus neutralization assay as described previously [4]”. Could author briefly summarize the assay in the Methods?
  • Could author include a brief description about how serum was harvested?
  • In the discussion, a speculation about how the situation may be after 3 doses of vaccine, as well as about the potential impact of the described results on the community could be included

Author Response

Dear Reviewer 1,

We wish to express our gratitude for your time and effort in reviewing our manuscript. Your comments helped us correct and improve the manuscript. Please refer to the point-by-point revisions we made in response to your comments.   

Reviewer 1:

In this manuscript, Estrada and colleagues performed a very relevant immunobridging study to compare the neutralizing antibody generated after vaccination with two doses of AstraZeneca and MVC-COV1901 vaccines. Overall, the results are very important for the scientific (and non-scientific) communities, and therefore they should be published.

Comments:

Please explain the concept of immunobridging already in the abstract

Thank you for this comment. Immunobridging has been explained on p.2 Lines 57-63. To further explain, we added the following sentences to the revised manuscript:

“ It is assumed that a part of the immune response – such as binding or neutralizing antibodies – will correlate with other important components of the immune response [4]. In the absence of efficacy data, immunobridging can be adopted as an approach to infer the likelihood of a vaccine’s protective effect, by translating immunogenicity to vaccine efficacy. The inference is based on the comparison of immunogenicity of a new vaccine with a comparator vaccine with an established protective effect [4].” 

It would be optimal to introduce the differences between the two types of vaccines (for example in the introduction, and eventually to discuss it in the discussion part)

Thank you for this comment. We included a comparison of the vaccines in the Methods section. Relevant text can be found on p. 3, Lines 127-133. In the discussion part, a comparison of the strength of immune response between two types of vaccines (i.e. adenoviral vector and protein subunit vaccines) was discussed briefly. This can be found in p. 10, Lines 332-339.

 “Results of the study corroborate earlier findings which suggest adenoviral vector vaccines generate less neutralizing antibodies but, not examined in this study, higher T cell responses compared to protein subunit vaccines [15]. For AZD1222, overall vaccine efficacy more than 14 days after the second dose was 66.7% [13]. Levels of neutralizing antibodies can be correlated and used to predict vaccine efficacy [16, 17]. Our findings suggest that levels of neutralizing antibodies of 2 doses of MVC-COV1901 correlate to approximately 90% vaccine efficacy against the prototype strain [17]. As mentioned above, this protein subunit vaccine also demonstrated better tolerability.”

The authors reported that individuals were generally healthy. Did they measure any objective parameter (for example CRP or inflammation in the blood) to define them healthy or not? In addition, were subjects who experienced COVID-19 before the current study excluded?

Thank you for the comment. Clinical laboratory tests were performed and vital signs were obtained at screening. Included In the sample were generally healthy individuals or those who have pre-existing medical conditions but are in a stable.A stable condition is defined as disease not requiring significant change in therapy or hospitalization for worsening disease 3 months before enrollment and expected to remain stable for the duration of the study As part of the exclusion criteria of both AZ and MVC studies, those with previous known or potential exposure to COVID-19 were excluded. This was written in p. 3, Lines 105-106.

In both studies, individuals with previous known or potential exposure to COVID-19 were excluded.”

Line 120: “Neutralizing antibody titres against the Wuhan prototype SARS-CoV-2 were determined by SARS-CoV-2 live virus neutralization assay as described previously [4]”. Could author briefly summarize the assay in the Methods?

Thank you for the comment. We further expanded the section on the assay. This is reflected in p. 4, Lines 164-179.

“Neutralizing antibody titers against the Wuhan prototype SARS-CoV-2 strain were determined by a live virus neutralization assay performed in our phase 2 clinical study. The detection and characterization of neutralizing antibodies were performed by central laboratories using validated live virus neutralization assays. The live SARS-CoV-2 neutralization assay was performed as previously described with wildtype SARS-CoV-2, Taiwan CDC strain number 4 (hCoV-19/ Taiwan/4/2020; GISAID accession ID EPI_ISL_411927) [6]. Briefly, t he serum samples underwent a total of eight two-fold dilutions, starting from a 1:8 dilution to a final dilution of 1:1024. Diluted serum samples were then mixed with an equal volume of 100 TCID50 per 50 μL of virus and incubated at 37°C for 1 hour. After incubation, the mixture was added to Vero E6 cells and incubated at 37°C in a 5% CO2 incubator for 4–5 days. The neutralizing titer (NT50) was estimated as the reciprocal of the highest dilution capable of inhibiting 50% of the cytopathic effect. The Reed-Muench method was used to calculate the NT50. Neutralizing antibody titers were then converted to the WHO Standardized Unit, IU/mL. The conversion was based on the WHO validated NIBSC reference panel. The GMT was derived from the results and converted to international units (IU/mL) [6].”

Could author include a brief description about how serum was harvested?

To address this comment, a sentence was added to briefly describe how serum was harvested. This sentence is written on p. 3 Lines 106-108.

“Serum for neutralizing antibody titers was collected on the day of vaccination (pre) and 28 days after the second dose of the vaccine (post).”

In the discussion, a speculation about how the situation may be after 3 doses of vaccine, as well as about the potential impact of the described results on the community could be included

Thank you for the comment. Findings of the study will have implications on the further development and approval of these vaccines. Results of previous studies on the MVC-COV1901 as a booster have proven that it elicits a robust immune response and has a favorable safety profile. In combination with the safety data, results of this immunobridging analysis support the use of protein subunit vaccines as safe and more tolerable options with a good immune response. These statements are written on p. 10, Lines 319-331.

“MVC-COV1901 was well tolerated without observed safety concerns in the phase 2 study [6] . In addition, the V-Watch program, launched by the Taiwan Centers for Disease Control to monitor post-marketing safety, has reported no serious adverse effects for the MVC-COV1901 vaccine.  The V-Watch database contains data from more than 2 million doses which have been administered as of March 2022. MVC-COV1901 presented a favorable safety profile compared to vaccines utilizing other platforms [12]. In combination with the safety data, results of this immunobridging analysis support the use of protein subunit vaccines as safe and more tolerable options with a robust immune response. Results of studies conducted to test MVC-COV1901 as a booster shot also warrant that the vaccine offers a robust immune response while maintaining a favorable safety profile [13,14]. These findings provide a safe and effective alternative for the completion of primary and booster immunization, which potentially can accelerate vaccination globally especially in low- and middle-income countries.” .”

Reviewer 2 Report

The paper describes a study performed in Taiwan, to compare neutralizing Ab levels at 28 days after second dose of vaccine in 903 subjects vaccinated with MVC-COV1901 and in a comparator group of 200 subjects previously vaccinated with AZD1222.

The 903 subjects vaccinated with MVC-COV1901 represent a subset (per protocol immunogenicity subset) of a previous phase 2 randomized trial (Hsieh et al, Lancet Respiratory Medicine, 2021, reference 4).

The authors report that the objective of the manuscript is to illustrate an example of a COVID-19 vaccine approved in Taiwan (according to the TFDA indications and criteria) using an  immunobridging study prior to the availability of efficacy data .

My main concern is related to the very weak study design (an external comparison rather than a RCT), as well as to the use of a surrogate endpoint (immunogenicity rather than infection/serious infection).

Below you will find some observations and comments that, in my opinion, could  improve the work in some sections:

- The authors should better clarify how the 200 previously vaccinated with AZD1222 were selected; were they selected ad hoc among those who came for vaccination with AZD1222, and were they asked to supply a blood sample before and 28 days after vaccination? Also, to better clarify the selection of 903 MVC-COV1901 subjects from the main study I suggest that the bibliographic reference (n.4) of the phase 2 trial published in Lancet Respiratory Medicine be added in the methods section (row 73).

- The authors should better specify in the methods section what sample size calculation they did for this non-inferiority study.

- In the method section, please specify for each sample the enrolment period and whether both samples included subjects who had not previously had Sars-COV2 infection

- The statistical analysis section should better specify which analyses were performed as sensitivity analysis

- Adjusted GMT ratios reported on page 6, line 192, differ from those reported in Table 3; please clarify

- In the discussion section, the limitations of the study should include the use of a surrogate endpoint (immunogenicity rather than infection/serious infection) and the limited safety evidence especially for rare events (based on 3800 patients included in the previous study).

- Tables 2 and 3 should be reformatted.

- English should be revised

Typos:

- page 3, line 114: Put a dot after "…external control group"

- page 4, first paragraph Immunogenicity: in subjects of under age of ..

- page 5, lines 163-165: probably this paragraph is part of the legend of Figure 1.

- page 8, line 224: perhaps the phrase should be " but used an external comparison" ?

- page 8, line 225: please check cell-medicated (mediated?)

Author Response

Dear Reviewer 2,

We wish to express our gratitude for your time and effort in reviewing our manuscript. Your comments helped us correct and improve the manuscript. Please refer to the point-by-point revisions we made in response to your comments.   

Reviewer 2:

The paper describes a study performed in Taiwan, to compare neutralizing Ab levels at 28 days after second dose of vaccine in 903 subjects vaccinated with MVC-COV1901 and in a comparator group of 200 subjects previously vaccinated with AZD1222.

The 903 subjects vaccinated with MVC-COV1901 represent a subset (per protocol immunogenicity subset) of a previous phase 2 randomized trial (Hsieh et al, Lancet Respiratory Medicine, 2021, reference 4).

The authors report that the objective of the manuscript is to illustrate an example of a COVID-19 vaccine approved in Taiwan (according to the TFDA indications and criteria) using an  immunobridging study prior to the availability of efficacy data .

My main concern is related to the very weak study design (an external comparison rather than a RCT), as well as to the use of a surrogate endpoint (immunogenicity rather than infection/serious infection).

Below you will find some observations and comments that, in my opinion, could  improve the work in some sections:

Comments:

The authors should better clarify how the 200 previously vaccinated with AZD1222 were selected; were they selected ad hoc among those who came for vaccination with AZD1222, and were they asked to supply a blood sample before and 28 days after vaccination? Also, to better clarify the selection of 903 MVC-COV1901 subjects from the main study I suggest that the bibliographic reference (n.4) of the phase 2 trial published in Lancet Respiratory Medicine be added in the methods section (row 73).

Participants for the AZD1222 trial were recruited among health care workers at Taoyuan General Hospital and administered two doses of AZ vaccine, 8 weeks apart. Revisions are reflected in p. 3 Line 100-104. Citation of the phase 2 trial was added in the sentence at p.3, Line 99.

 The authors should better specify in the methods section what sample size calculation they did or this non-inferiority study.

Thank you for the comment. The samples came from 2 different studies and sample sizes were determined independently. For the MVC-COV1901 trial, the sample size was based on lot-to-lot consistency. The estimation is based on two sided 95% CI [100(1-2α)%] calculated using the differences of the post-vaccination GMT of neutralizing antibody titers between a pair of lots. The total sample size was based on the minimum requirement of the Taiwan Food and Drugs Administration (TFDA). As for the AZD1222 study, there were no data available to compute for the sample size hence it was not justified at the time the protocol was written. This explanation was written on p.3, Lines 109-116.

- In the method section, please specify for each sample the enrolment period and whether both samples included subjects who had not previously had Sars-COV2 infection

Thank you for this comment. As part of the exclusion criteria of both AZ and MVC studies, those with previous known or potential exposure to COVID-19 were excluded. This was written in p. 3, Lines 105-106.

“In both studies, individuals with previous known or potential exposure to COVID-19 were excluded.”

- The statistical analysis section should better specify which analyses were performed as sensitivity analysis

Thank you for the comment. Sensitivity analyses have been further described on p. 4, Lines 187-198.

“The GMTs are presented with their two-sided 95% CIs. Sensitivity analyses were performed by excluding elderly participants and those with elevated baseline neutralizing antibody levels. The most extreme modification of the dataset excluded participants with a GMT higher than the 67th percentile at Day 57. Assessment of the magnitudes of differences in immune response between the two vaccines was conducted using an analysis of covariance (ANCOVA) model. The model included the log-transformed antibody titers at Day 57 as the dependent variable, vaccine group (AZD1222 and MVC-COV1901) as an explanatory variable and adjusted for age, BMI, gender and comorbidity profile. The 95% CI for the adjusted neutralizing antibody titers of each vaccine group were obtained. Then adjusted GMT and corresponding 95% CI were back-transformed to the original scale.”

- Adjusted GMT ratios reported on page 6, line 192, differ from those reported in Table 3; please clarify

Relevant section was revised. The estimate was omitted as it’s already reflected in table 3 (p.8).

- In the discussion section, the limitations of the study should include the use of a surrogate endpoint (immunogenicity rather than infection/serious infection) and the limited safety evidence especially for rare events (based on 3800 patients included in the previous study).

Thank you for your comment. The use of immunogenicity is for a lack of efficacy data or information on infection/serious infection. Immunobridging permits the comparison of a vaccine with no efficacy data to one with known and established efficacy through the correlation of antibody titers or other important components of the immune response. We therefore consider the use of immunogenicity not as a limitation of the study but as the main feature of the immunobridging approach. For the safety profile, the V-Watch program, launched by the Taiwan Centers for Disease Control to monitor the safety post-vaccination, reports no serious adverse effects for MVC-COV1901. This database contains data from more than 2 million doses which have been administered as of March 2022. This information was cited on p. 10, Lines 320-324.

- Tables 2 and 3 should be reformatted.

Tables were revised. Edited tables are on page 8.

Table 2. Neutralizing antibody titers and adjusted GMT ratios in subjects immunized with either two doses of AZD1222 or MVC-COV1901 in all ages and ages 20-64 years at Day 57 (28 days after the second dose).

Item\MVC lot

<65 years

All ages

AZD1222

N = 192

MVC-COV1901

(PPI subset)

N = 674

p-value

(GMT ratio)

AZD1222

N = 198

MVC-COV1901

(PPI subset)

N = 893

p-value

(GMT ratio)

GMT

(95% CI)

185.97
(167.3 ~ 206.7)

723.6

(683.7 ~ 765.8)

184.05

(166.5 ~ 204.7)

654.07

(620.9 ~ 689.03)

   GMT Ratio of MVC/AZ

(95% CI)

3.89

(3.45 ~ 4.4)

<0.0001

3.55

(3.2 ~ 3.97)

<0.0001

   Adjusted GMT Ratio*

(95% CI)

3.78

(3.3 ~ 4.3)

<0.0001

3.8

(3.4 ~ 4.3)

<0.0001

*The GMT Ratio was adjusted for age, gender, BMI, and comorbidity profile using general linear models.

GMT- Geometric Mean Titer, PPI – Per Protocol Immunogenicity

Table 3. Sensitivity analysis excluding subjects with a GMT higher than the 67th percentile of neutralizing antibody titer in subjects immunized with either two doses of AZD1222 or MVC-COV1901 in all ages and ages 20-64 years at Day 57 (PPI subset).

Item\MVC lot

<65 years

All ages

AstraZeneca
AZD1222

N = 192

MVC-COV1901

(PPI subset)

N = 458

p-value

(GMT ratio)

AstraZeneca
AZD1222

N = 198

MVC-COV1901

(PPI subset)

N = 635

p-value

(GMT ratio)

GMT

(95% CI)

185.97
(167.3 ~ 206.7)

492.6

(470.1 ~ 516.1)

184.05

(166.5 ~ 204.7)

453.7

(433.8 ~ 474.7)

   GMT Ratio of MVC/AZ

(95% CI)

2.65

(2.4 ~ 2.97)

<0.0001

2.46

(2.2 ~ 2.7)

<0.0001

   Adjusted GMT Ratio*

(95% CI)

2.6

(2.3 ~ 2.9)

<0.0001

2.62

(2.3 ~ 2.9)

<0.0001

*The GMT Ratio was adjusted for age, gender, BMI, and comorbidity profile using general linear models.

GMT- Geometric Mean Titer, PPI – Per Protocol Immunogenicity

English should be revised

 Thank you for your comment. The manuscript was reviewed and edited to improve the English language and writing style.

Typos:

- page 3, line 114: Put a dot after "…external control group"

This was corrected on p. 4 Line 158.

- page 4, first paragraph Immunogenicity: in subjects of under age of ..

This was corrected on p.6, Line 225.

- page 5, lines 163-165: probably this paragraph is part of the legend of Figure 1.

This was corrected. It is the caption for the figure in p.6.

- page 8, line 224: perhaps the phrase should be " but used an external comparison" ?

The phrase was omitted.

- page 8, line 225: please check cell-medicated (mediated?)

This was corrected on p.10, Line 349.

Reviewer 3 Report

This paper considers the efficacy of vaccines as an important issue as a comparative analysis with previously evaluated and approved vaccines.

1. Please correct the location of Page 1. Line 28, (AZD1222)
2.Page 2.Line 87, Medigen COVID-19 vaccine is the first to appear, but there is no specific explanation. Additional explanation is required.
3. Page 2, line 74-75, line 90, ChAdOx74nCOV-19 (AZD1222), and ChAdOxAZD1222 have different terms.
4. You need to explain the purchase or sale of Page 3, line 120, and Wuhan prototype SARS-CoV-2, registration number, etc.
5. Methods does not describe how to sustain results on Page 6, line 175, and results. It is necessary to improve the understanding of the study by describing the method.
6. The structure of Page 6, line 195, and table is not clear. Please revise and arrange it.
7.Page 8, whether national vaccine permits can be granted with the limitations mentioned by the authors? Is the limit mentioned because it is the limit because it differs from the existing vaccine permit? Can I get a vaccine license if I study and get the results from this paper? I'd appreciate it if you could explain more.

Author Response

Dear Reviewer 3,

We wish to express our gratitude for your time and effort in reviewing our manuscript. Your comments helped us correct and improve the manuscript. Please refer to the point-by-point revisions we made in response to your comments.   

Reviewer 3

This paper considers the efficacy of vaccines as an important issue as a comparative analysis with previously evaluated and approved vaccines.

Comments:

Please correct the location of Page 1. Line 28, (AZD1222)

Thank you for the comment. This was corrected on p. 1, Line 33.

Page 2.Line 87, Medigen COVID-19 vaccine is the first to appear, but there is no specific explanation. Additional explanation is required.

Thank you for the comment. MVC-COV1019 is now described on p.3, Lines 127-129.

“MVC-COV1901 is, a subunit vaccine consisting of the SARS-CoV-2 prefusion spike protein (S-2P) adjuvanted with 750 μg CpG 1018 and 375 μg aluminum hydroxide. A standard 0.5 mL dose contains 15 µg of the Spike-2P.”

Page 2, line 74-75, line 90, ChAdOx74nCOV-19 (AZD1222), and ChAdOxAZD1222 have different terms.

Corrected section can be found on p.3, Line 130.  

You need to explain the purchase or sale of Page 3, line 120, and Wuhan prototype SARS-CoV-2, registration number, etc.

Thank you for the comment. We further expanded the section on the assay. This is reflected in p. 4, Lines 164-179.

“Neutralizing antibody titers against the Wuhan prototype SARS-CoV-2 strain were determined by a live virus neutralization assay performed in our phase 2 clinical study. The detection and characterization of neutralizing antibodies were performed by central laboratories using validated live virus neutralization assays.The live-SARS-CoV-2 neutralization assay was performed as previously described with wildtype SARS-CoV-2, Taiwan CDC strain number 4 (hCoV-19/ Taiwan/4/2020; GISAID accession ID EPI_ISL_411927) [6]. Briefly, the serum samples underwent a total of eight two-fold dilutions, starting from a 1:8 dilution to a final dilution of 1:1024. Diluted serum samples were then mixed with an equal volume of 100 TCID50 per 50 μL of virus and incubated at 37°C for 1 hour. After incubation, the mixture was added to Vero E6 cells and incubated at 37°C in a 5% CO2 incubator for 4–5 days. The neutralizing titer (NT50) was estimated as the reciprocal of the highest dilution capable of inhibiting 50% of the cytopathic effect. The Reed-Muench method was used to calculate the NT50. Neutralizing antibody titers were then converted to the WHO Standardized Unit, IU/mL. The conversion was based on the WHO validated NIBSC reference panel. The GMT was derived from the results and converted to international units (IU/mL) [6].”

Methods does not describe how to sustain results on Page 6, line 175, and results. It is necessary to improve the understanding of the study by describing the method.

Thank you for the comment. We further explained the reverse cumulative distribution curve (RCDC). We explained that RCD curves are step functions based on the order statistics of the data. Revisions are written on pp. 5, Lines 200-207.

“Lastly, to illustrate the strength of the immune response, reverse cumulative distribution (RCD) curves were constructed for data 28 days after the 2nd dose for the AZD1222 and MVC-COV1901 groups. As described by Reed et al. [10], RCD curves are step functions based on the order statistics of the data. The curve begins with a value of 1.0 or 100% at an antibody titer of zero and falls to a value of zero above the highest titer. In the case of ties, the step size becomes the equivalent of the number of tied values times 1/n [10].”

The structure of Page 6, line 195, and table is not clear. Please revise and arrange it.

Tables were revised. Edited tables are on page 8.

Table 2. Neutralizing antibody titers and adjusted GMT ratios in subjects immunized with either two doses of AZD1222 or MVC-COV1901 in all ages and ages 20-64 years at Day 57 (28 days after the second dose).

Item\MVC lot

<65 years

All ages

AZD1222

N = 192

MVC-COV1901

(PPI subset)

N = 674

p-value

(GMT ratio)

AZD1222

N = 198

MVC-COV1901

(PPI subset)

N = 893

p-value

(GMT ratio)

GMT

(95% CI)

185.97
(167.3 ~ 206.7)

723.6

(683.7 ~ 765.8)

184.05

(166.5 ~ 204.7)

654.07

(620.9 ~ 689.03)

   GMT Ratio of MVC/AZ

(95% CI)

3.89

(3.45 ~ 4.4)

<0.0001

3.55

(3.2 ~ 3.97)

<0.0001

   Adjusted GMT Ratio*

(95% CI)

3.78

(3.3 ~ 4.3)

<0.0001

3.8

(3.4 ~ 4.3)

<0.0001

*The GMT Ratio was adjusted for age, gender, BMI, and comorbidity profile using general linear models.

GMT- Geometric Mean Titer, PPI – Per Protocol Immunogenicity

Table 3. Sensitivity analysis excluding subjects with a GMT higher than the 67th percentile of neutralizing antibody titer in subjects immunized with either two doses of AZD1222 or MVC-COV1901 in all ages and ages 20-64 years at Day 57 (PPI subset).

Item\MVC lot

<65 years

All ages

AstraZeneca
AZD1222

N = 192

MVC-COV1901

(PPI subset)

N = 458

p-value

(GMT ratio)

AstraZeneca
AZD1222

N = 198

MVC-COV1901

(PPI subset)

N = 635

p-value

(GMT ratio)

GMT

(95% CI)

185.97
(167.3 ~ 206.7)

492.6

(470.1 ~ 516.1)

184.05

(166.5 ~ 204.7)

453.7

(433.8 ~ 474.7)

   GMT Ratio of MVC/AZ

(95% CI)

2.65

(2.4 ~ 2.97)

<0.0001

2.46

(2.2 ~ 2.7)

<0.0001

   Adjusted GMT Ratio*

(95% CI)

2.6

(2.3 ~ 2.9)

<0.0001

2.62

(2.3 ~ 2.9)

<0.0001

*The GMT Ratio was adjusted for age, gender, BMI, and comorbidity profile using general linear models.

GMT- Geometric Mean Titer, PPI – Per Protocol Immunogenicity

Page 8, whether national vaccine permits can be granted with the limitations mentioned by the authors? Is the limit mentioned because it is the limit because it differs from the existing vaccine permit? Can I get a vaccine license if I study and get the results from this paper? I'd appreciate it if you could explain more.

The Medigen COVID-19 vaccine EUA was granted on Jul. 30, 2021 in Taiwan, and subsequently more than 300,000 adults were vaccinated with the vaccine. Indeed, most licensed COVID-19 vaccines underwent phase 3 study in the era of the Wuhan and Alpha variants, but immunobridging study is to evaluate neutralizing antibody, compared with licensed COVDI-19 vaccines which completed phase 3 study. However, more and more mutant variants of SARS CoV-2 were detected, and different variants could induce different neutralizing antibody. Medigen COVID-19 vaccine had been granted on Jul. 30, 2021 in Taiwan.